# Antivirals for Broader Coverage against Human Coronaviruses

**DOI:** 10.3390/v16010156

**Published:** 2024-01-20

**Authors:** Mia Outteridge, Christine M. Nunn, Kevin Devine, Bhaven Patel, Gary R. McLean

**Affiliations:** 1School of Human Sciences, London Metropolitan University, London N7 8DB, UK; mio0358@my.londonmet.ac.uk (M.O.); c.nunn@londonmet.ac.uk (C.M.N.); k.devine@londonmet.ac.uk (K.D.); b.patel1@londonmet.ac.uk (B.P.); 2National Heart and Lung Institute, Imperial College London, London W2 1PG, UK

**Keywords:** coronavirus, antiviral, biologic

## Abstract

Coronaviruses (CoVs) are enveloped positive-sense single-stranded RNA viruses with a genome that is 27–31 kbases in length. Critical genes include the spike (S), envelope (E), membrane (M), nucleocapsid (N) and nine accessory open reading frames encoding for non-structural proteins (NSPs) that have multiple roles in the replication cycle and immune evasion (1). There are seven known human CoVs that most likely appeared after zoonotic transfer, the most recent being SARS-CoV-2, responsible for the COVID-19 pandemic. Antivirals that have been approved by the FDA for use against COVID-19 such as Paxlovid can target and successfully inhibit the main protease (MPro) activity of multiple human CoVs; however, alternative proteomes encoded by CoV genomes have a closer genetic similarity to each other, suggesting that antivirals could be developed now that target future CoVs. New zoonotic introductions of CoVs to humans are inevitable and unpredictable. Therefore, new antivirals are required to control not only the next human CoV outbreak but also the four common human CoVs (229E, OC43, NL63, HKU1) that circulate frequently and to contain sporadic outbreaks of the severe human CoVs (SARS-CoV, MERS and SARS-CoV-2). The current study found that emerging antiviral drugs, such as Paxlovid, could target other CoVs, but only SARS-CoV-2 is known to be targeted in vivo. Other drugs which have the potential to target other human CoVs are still within clinical trials and are not yet available for public use. Monoclonal antibody (mAb) treatment and vaccines for SARS-CoV-2 can reduce mortality and hospitalisation rates; however, they target the Spike protein whose sequence mutates frequently and drifts. Spike is also not applicable for targeting other HCoVs as these are not well-conserved sequences among human CoVs. Thus, there is a need for readily available treatments globally that target all seven human CoVs and improve the preparedness for inevitable future outbreaks. Here, we discuss antiviral research, contributing to the control of common and severe CoV replication and transmission, including the current SARS-CoV-2 outbreak. The aim was to identify common features of CoVs for antivirals, biologics and vaccines that could reduce the scientific, political, economic and public health strain caused by CoV outbreaks now and in the future.

## 1. Introduction

### 1.1. History of Human Coronaviruses

The 2019 Human Coronavirus (COVID-19) outbreak, caused by Severe Acute Respiratory Syndrome Coronavirus 2 (SARS-CoV-2), has caused public health, economic and political devastation on a global scale [1,2]. Human coronaviruses (CoVs) are single-stranded, positive-sense RNA viruses, with genomes ranging between 27,000 and 31,000 bases (b) in length [3]. The CoVs are transmitted via aerosols from infected individuals and by direct contact with contaminated surfaces, which can be prevented by handwashing with soap, social distancing and utilising personal protective equipment (PPE) [4]. There are currently seven CoVs that infect humans, with the first identification in the 1960s and the most recent in 2019, although new variants of SARS-CoV-2 are still appearing such as Omicron, which has frequently emerging subvariants including Pirola (BA.2.86) [5].

CoV-229E and CoV-OC43 were the first CoVs identified in the 1960s [2], while the first severe CoV, known as SARS-CoV, was identified in 2003 in China [6]. CoV-NL63 and CoV-HKU1 were detected as a result of increased testing following the SARS-CoV outbreak [7]. CoV-229E, CoV-OC43, CoV-NL63 and CoV-HKU1 are self-limiting infections of the upper respiratory tract and present with mild common-cold-like symptoms, whilst SARS-CoV, MERS-CoV and SARS-CoV-2 are responsible for increased clinical severity by infecting the lower respiratory tract, leading to complications, such as pneumonia and other severe lung pathology [8].

In 2003, in Guangdong, China, SARS-CoV was the first severe HCoV to be identified. It caused acute respiratory distress syndrome (ARDS), pneumonia and even respiratory failure, as well as other complications, such as liver or kidney impairment and diastolic cardiac dysfunction [9,10]. Nosocomial infections and transmission from very sick symptomatic individuals tended to follow within the immunocompromised and elderly populations [6]. However, whilst spreading across >30 different countries, ~83% of all 8096 laboratory-confirmed cases remained in China, with 774 deaths worldwide as of July 2003 and a case fatality rate of approximately 9.5% [6]. The first cases were documented from November 2002, and the SARS epidemic continued until June 2003, with a brief peak in April 2003, but a better understanding of control measures was implemented by then, and the cases decreased, with none having been documented since 2004 [11].

Middle Eastern Respiratory Syndrome CoV (MERS-CoV) is another severe CoV identified in Saudi Arabia in 2012, with camels being the known reservoir [12]. Transmission largely occurs directly from camels to humans, with occasional human-to-human transmission from very close contact with infected individuals. Between April 2012 and June 2023, there were 2605 laboratory-confirmed cases of MERS-CoV, with 936 associated deaths and 36% case–fatality ratio [13]. Most of these cases were identified in Saudi Arabia that had 2196 cases and 855 deaths in total [13]. Currently, ~40% of cases are nosocomial, potentially due to overcrowded hospitals and co-morbidities increasing infection susceptibility [14]. In fact, MERS is only very rarely transmitted between people outside of hospital settings and is, therefore, considered low risk for the general population [15].

SARS-CoV-2 is the most recent severe HCoV, which emerged in late 2019 in Wuhan, China, and subsequently spread globally, with >700 million confirmed cases and >6.95 million deaths [16]. SARS-CoV-2 has been shown to have a higher transmission rate (reproductive number: 2.9; however, in other studies, it has been demonstrated that it can be anywhere between 2.6 and 4.71) than SARS-CoV (reproductive number: 1.77) but, interestingly, has a much lower fatality rate when measured against its infection rate. SARS-CoV had a case fatality rate of 9.5% and SARS-CoV-2 was 2.13% [17,18].

The severe human CoV’s are thought to have originated from bats [19]. Bats harbour the most zoonotic diseases compared to other mammalian taxonomic groups and are one of the most abundant mammalian orders known, comprising 20% of biodiversity [20]. Viruses, such as filoviruses, paramyxoviruses and coronaviruses, show evidence of emerging from various bat species, which may be because of the long lifespan of bats (20–40 years), worldwide distribution and high metabolic rates. Research shows that bats also have immunological characteristics that control virus propagation [20,21,22]. Viruses generally have very high evolution rates, especially RNA viruses, because of their rapid mutation rates and instable genetic heterogeneity. This is compounded by high host population density and short generation times [23]. SARS-CoV-2 has a much higher evolutionary rate compared to the other HCoVs, which is reflected in numerous mutations within the Spike gene that have not mutated in others; therefore, the unpredictability of the different CoV strains’ evolution creates difficulty in preparing for outbreaks and, ultimately, in providing control measures. However, this may also be a result of predominantly sequencing SARS-CoV-2 compared to other CoVs because of the urgency at the time [20,22,24].

### 1.2. CoV Genes and Genetic Diversity

SARS-CoV and SARS-CoV-2 are ~80% genetically similar to each other, providing some evidence that the virus evolved and re-emerged from their original reservoirs, with epidemiological and genetic studies indicating that SARS-CoV and SARS-CoV-2 originated from the BANAL coronaviruses found in horseshoe (*Rhinolophus*) bats in Laos [25,26]. Figure 1 presents the phylogenetic classes representing the genetic differences between SARS-CoV-2 and other HCoVs by tracking mutagenic shifts in their genomic sequences using FigTree v1.4.4 [27,28,29].

Figure 1 shows that CoV-NL63 and CoV-229E have the closest similarity to each other, CoV-OC43 with CoV-HKU1 and MERS-CoV with SARS-CoV-1 and SARS-CoV-2 based on their genomic sequences. Bats and rodents are known to be the primary reservoirs for all seven human CoVs by using epidemiological data to track and trace the index, primary and secondary cases through human migration [33]. SARS-CoV-2 has ~96.8% genetic similarity to BANAL-52 CoV found in the *R. malayanus* species of horseshoe bats in Laos [34], therefore providing evidence of bats generally being the original reservoir for severe CoVs. The CoVs are categorised in different genera based on their genetic similarity and evolutionary patterns, as displayed in Figure 2. The α-CoV and ß-CoV include the seven CoVs that infect humans, whilst γ-CoV and δ-CoV predominantly consist of rodent and avian CoVs [29].

The human CoV genomes have a typical structure containing multiple open reading frames (ORFs) that encode non-structural proteins (NSPs) 1–16, which are responsible for viral replication, virion formation and immune evasion [36]. This arrangement is shared by the seven human CoVs, with subtle differences found at the 3′ end of the genome, where the viral structural protein genes and accessory ORFs are found. Further details of the SARS-CoV-2 schematics are shown in Figure 3 and Table 1 [36].

ORF1ab is the largest section of the genome, which encodes for polyproteins 1a and 1b (pp1ab) [37], pp1a encodes for NSPs 1–11 and pp1b encodes for NSPs 12–16, which are made as long polypeptides before further processing [38]. A detailed summary of the different viral proteins is presented in Table 1.

**Table 1 viruses-16-00156-t001:** A table summarising the different SARS-CoV-2 proteins including the S, E, M and N proteins, as well as the different ORFs/NSPs and their functions where known. PDB codes of known protein structure are shown.

Protein	Functions in SARS-CoV-2	3D Structure
NSP1	~180 a.a. promotes viral gene expression and inhibits immune functions [39].	7K7P
NSP2	~638 a.a. interacts with host factors prohibitin 1 and prohibitin 2, which are involved in many cellular processes including mitochondrial biogenesis [40].	7EXM (N terminus)
NSP3	~1945 a.a, papain-like protease (PLpro) processes the viral polyprotein [41].	6WOJ
NSP4	~500 a.a. transmembrane glycoprotein forms DMVs in complex with NSP3 [42].	Not available
NSP5	~306 a.a, main protease (3CLpro/MPro) cleaves polyprotein release nsp4-nsp16 [43].	8DDI
NSP6	~290 a.a, complexes with NSP3 and NSP4 to induce DMVs in infected cells [44].	8DRS (complex with NSP6 and 7)
NSP7	~83 a.a. forms supercomplex with NSP8 and NSP12 to process viral RNA [45].	7DCD (complex with NSP8)
NSP8	~198 a.a forms supercomplex with NSP7 and NSP12 to process viral RNA [45].	7DCD (complex with NSP7)
NSP9	~113 a.a interacts with NSP12, unclear specific functions [46].	3EE7
NSP10	~139 a.a. forms a dodecamer complex with NSP14 and NSP16 to stimulate 3′-5′ exoribonuclease and 2′-O-methyltransferase activities [47].	6W4H (complex with NSP16)
NSP11	~13–23 a.a pp1a cleavage product at the nsp10/11 boundary. Function unknown [48].	Not available
NSP12	~932 a.a RdRp performing replication and transcription of the viral genome [49].	7BW4
NSP13	~601 a.a the main helicase for the CoVs [50].	7NIO
NSP14	~527 a.a 3′-5′ exoribonuclease proofreading mechanism (ExoN) in complex with NSP10 and viral mRNA capping activities [51].	7QGI
NSP15	~346 a.a important for immune evasion by preventing dsRNA sensor activation [52].	7DW0 (complex with NSP5/14)
NSP16	~298 a.a 2′-O-methyltransferase activity, activated once in complex with NSP10 [53].	6W4H (complex with NSP10)
ORF3a	~275 a.a viroporin iron channel in SARS-CoV, promotes viral movement, release and activation of inflammasomes [54].	6XDC
ORF3b	~22 a.a interrupting interferon antagonistic functions, but not fully understood [55].	Not available
ORF6	~61 a.a interferes with innate immune responses, suppressing kinases and types I and II IFN pathways [56].	7VPH (complex with Rae1-Nup98)
ORF7a	~121 a.a type I membrane protein that interacts with CD14+ monocytes and increases glycosylation for immune evasion of presenting antigens [57].	7CI3 (ectodomain)
ORF7b	~43 a.a interference with cellular processes and infection symptoms [57].	Not available
ORF8	~121 a.a interferon antagonist to promoting cytokine storms [57].	7JTL
ORF9b	~97 a.a. localised in mitochondrial membranes, associated with IFN responses [58].	7YE7
ORF9c	~70 a.a interacts with host proteins, involvement in ER stress responses and lipid remodelling [57].	Not available
ORF10	~38 a.a not necessary for SARS-CoV-2 infection [57].	Not available
Spike (S) (ORF2)	~1273 a.a interacts with host entry receptors, facilitates fusion [59].	8C8P (complex with mAb).
Membrane (M) (ORF5)	~222 a.a mediates assembly, packaging and budding of viral particles [60]	8CTK
Envelope (E) (ORF4)	~75 a.a involved in viral assembly, budding, and pathogenesis [61].	7K3G
Nucleocapsid (N) (ORF9a)	~419 a.a. involved in genome protection, viral RNA replication, virion assembly, and immune evasion [62].	6WZO

Each viral protein has specific functions, which contribute to viral replication. For example, NSP3 is a protease, which cleaves and divides NSP1, 2 and 3 into individual components, while NSP5 is the main protease (MPro), which cleaves the remaining NSPs [4,5,6,7,8,9,10,11,12,13,14,15,16,63]. NSP12 is an RNA-dependent RNA polymerase (RdRp), and it is the central reservoir for RNA transcription and elongation, making it a critical NSP for viral replication [64]. RNA viruses normally have low RdRp fidelity, resulting in high mutations; however, HCoVs have an exonucleolytic proofreading mechanism regulated by NSP14 [65,66,67]. Favourable single nucleotide polymorphisms (SNPs) still occur to improve infectivity and immune evasion, negatively impacting the development of suitable treatments [67]. Most HCoVs possess almost all the genes/proteins in Table 1; however, the betacoronaviruses (CoV-HKU1 and CoV-OC43) also have genes that encode the haemagglutinin-esterase (HE) protein. HE contains a receptor-destroying sialate-O-acetylesterase domain and a receptor-binding lectin domain for O-Ac-Siac, which contributes to virion attachment and the breakdown of sialoglycotopes [68].

The Spike protein binds to the angiotensin-converting enzyme 2 (ACE2) protein found on the surface of host cells, which allows for fusion between the cell membrane and viral envelope. The nucleocapsid protein sheds viral RNA into the cytoplasm to form replication complexes [38,69,70]. The envelope, membrane, Spike and nucleocapsid protein translations are supported by double-membrane vesicles to form replication–transcription complexes [69]. After synthesis at the ER and Golgi apparatus, the proteins form into virions for budding and exocytosis to infect surrounding cells [55]. The final structure of the virus is presented in Figure 4.

The Spike protein is a common target for vaccine research due to its critical nature in virus entry and due to being the major target of the immune response to infection [59,73]. It is divided into subunit 1 (S1), which contains the receptor-binding domain (RBD) that interacts with the entry receptors [38], and subunit 2 (S2) that facilitates cellular fusion for viral entry [74]. The membrane protein is the most abundant in the viral particle, and it forms protein–RNA complexes for lipid bilayer stability [75]. The nucleocapsid protein is critical for stabilising and shielding the genome and facilitates exocytosis and immunogenic properties, making it a vaccine and diagnostic target [76]. The envelope protein is responsible for viral budding and envelope formation [77]. More details of each structural protein function can be found in Table 1. Most mutations amongst the HCoVs occur in the final third of their genomes, and mutations within the RBD of the Spike protein have resulted in the HCoVs evolving to bind different cell entry receptors; these are presented in Table 2 in Section 1.3.

### 1.3. CoV Genetic Drift

Mutations are defined as a change in an organism’s genetic sequence, which can be an insertion, deletion or point mutation, which does not always have a functional effect [78]. They can occur in both animal and human reservoirs. Multiple animals of different species can carry a progenitor SARS-CoV-2 and potentially infect humans in different locations, which can further impact how viruses may mutate considering population sizes [79].

Viral mutations often occur because of RNA polymerase instability and low fidelity [80]. CoVs possess an exoribonucleolytic proofreading mechanism (NSP14) that maintains its long genome and protects from frequent mutations [23]. Viruses with a longer genome are less prone to sporadic mutations because of the evolutionary addition of this proofreading gene; simpler viruses with smaller genomes, such as influenza, tend to mutate at much higher rates [81]. Very high population rates combined with high levels of sporadic mutations lead to a plethora of different variants, which will be harder to control immunologically [82].

Even though CoVs have this proofreading mechanism, there is evidence of multiple evolutionary mutations within its Spike protein, which are responsible for the direct binding and entrance of viral material into host cells [83]. A very small portion of mutations are expected to be impactful on the viral phenotype, which will positively influence infectivity, pathogenicity and transmissibility [84]. Studies on 229E-CoV and NL63-CoV have shown that there is no correlation between phenotypic evolution and the mutation rate; therefore, mutations are rare, with little evolutionary pressure [85,86]. The mutations that occur in the Spike protein can improve transmissibility and immune evasion, and this infers that mutations that happen here will have the same effect in all HCoVs [87].

The areas of a viral genome have different mutation rates that influence viral fitness for new hosts and are evidence of ongoing evolution. Studies have shown that the mutations that were most prominent in the COVID-19 pandemic were within the Spike (202 genomes had 34 Spike mutations, D614G being most frequent in 160 genomes) and nucleocapsid (65 genomes had 25 variations, R203K being the most frequent in 21 genomes) [88]. The receptor-binding motif (RBM) is a ‘hotspot’ of unique mutations. In April 2020, the D614G mutation was highly prominent and contributed to improved infectivity/transmissibility, and other Spike mutations resulted in the emergence of multiple variants of SARS-CoV-2, such as alpha, beta, delta and omicron [89,90]. Because of the fast rate of Spike mutations, it would not be considered an effective antiviral target for multiple CoVs. For example, there were ~>30 mutations in the Spike protein in the emergent omicron SARS-CoV-2 strain, which greatly increased its infectivity and improved antiviral and immune evasion [91].

Genomic sequence changes or mutations can cause functional and structural changes to proteins. Even as little as one amino acid placement can completely change a protein’s structure and function [92]. 229E-CoV, similar to SARS-CoV-2, has significant variability in the Spike and nucleocapsid proteins, most of which are found within the Spike RBD and affect its binding capability to host cells [85]. In January 2022, two novel OC43-CoV variants emerged, and it was found that even though the critical sialoglycan receptor-interacting residues were conserved, there were significant sequence mutations in other areas of the receptor-binding motif across these OC43-CoV isolates [93,94]. Human CoVs are clearly adaptable and open to selective pressure as the seven HCoVs have several entry pathways and bind to different host receptors (Table 2).

**Table 2 viruses-16-00156-t002:** The entry receptors that each of the HCoVs bind to for host cellular entry [95,96].

HCoV	Entry Receptors
CoV-NL63	Angiotensin converting enzyme 2 (ACE-2)
CoV-229E	Aminopeptidase N (APN)
CoV-HKU1	9-*O*-acetylated sialic acids (9-*O*-Ac-Sias)
CoV-OC43	9-*O*-acetylated sialic acids (9-*O*-Ac-Sias)
SARS-CoV	Angiotensin converting enzyme 2 (ACE-2)
MERS-CoV	Dipeptidyl peptidase 4 (DPP4)
SARS-CoV-2	Angiotensin converting enzyme 2 (ACE-2)

ACE2 is the cell entry receptor for NL63-CoV, SARS-CoV and SARS-CoV-2. NL63-CoV Spike protein has a very low sequence similarity compared to the severe beta CoVs (SARS-CoV: 23.7% and SARS-CoV-2: 25%), indicating significant divergence; however, they all have a conserved glycine residue (NL63-CoV: G537, SARS-CoV: G488 and SARS-CoV-2: G502), which is essential for ACE2 binding. In studies conducted by Rawat, P., et al., 2020, it was revealed, during bioinformatic analysis using FoldX and CUPSTAT, that the mutation of the conserved glycine residue destabilises the protein, and interaction with ACE2 is hindered, proving that it is important for binding [97]. Once bound to ACE2, acid-dependent proteolytic cleavage occurs in the S1 unit by proteases, such as the transmembrane protease serine proteases (TMPRSS), cathepsins and human airway trypsin-like proteases [98]. Spike cleavage exposes the fusion peptides of subunit 2 (S2); subsequently, the virus fuses with the host cell entry receptor of the cellular membrane [98].

CoV-OC43 and CoV-HKU1 use cellular glycocalyx sialylated compounds to enter host cells [99]. The Spike protein binds to sugar-based receptor determinants such as 9-O-acetylated sialic acids, and the haemaglutin-esterase (HE) that these betacoronaviruses possess is an acetylesterase, with a 9-O-acetylated sialic acid-specific lectin domain, which acts as a receptor and, therefore, breaks down the complex for viral entry [100]. Aminopeptidase N is the cell entry receptor for CoV-229E, also called CD13, which resides in fibroblastic epithelial cells of the lungs [101]. MERS-CoV utilises the dipeptidyl peptidase 4 (DPP4) cell entry receptor. Once the Spike protein has undergone proteolytic activation via the TMPRSS2 or cathepsin L pathways, residues in S1 can directly bind to DPP4 [102].

## 2. Treatment of HCoV Infection

### Antivirals

Antivirals are a class of small molecules that are designed to specifically target and interfere with viral replication by targeting key stages in the viral life cycle, such as viral attachment, uncoating, genome replication/translation or budding [103,104,105]. Antivirals target many mechanisms that target viral enzymes, such as proteases, polymerases or integrases, or they may be able to target viral surface membrane proteins, such as envelope or glycoproteins, the aim being to indirectly hinder the particular viral functions by allosterically altering enzymatic active sites and interrupting replication [106,107]. Some antivirals are designed without a 3′ hydroxyl group, which blocks the viral chain and prevents DNA/RNA synthesis and elongation [108]. Other inhibitor targets, for example, HIV protease inhibitors, will commonly involve a reaction between a hydroxyl group from the inhibitor and a carboxyl group of the protease active site (Gly27, Gly48, Asp29 and Asp30 are conserved in many HIV strains, so these are common amino acid targets) [109]. HIV protease inhibitors such as ritonavir also inhibit cytochrome (CYP) P450 enzymes that can be used for SARS-CoV-2; it increases the half-life of nirmatrelvir, the MPro inhibitor of SARS-CoV-2 [110]. Antiviral activity is achieved by either competitive binding to the substrate or by non-competitively binding to the enzyme to alter its 3D configurational shape [111,112]. Examples of different drugs and their targets are detailed below.

Artificially derived nucleoside analogue antivirals resemble naturally produced nucleosides that bind to a target site on viral RNA [113]. By binding to a viral enzyme involved in replication, the nucleosides will be converted into an active form, nucleotide, and host cells will transcribe the nucleoside analogue as if it was part of the host’s natural make-up [113]. As a result, target enzymes cannot function normally, and this interrupts their viral lifecycle [114].

Favipiravir is a guanosine nucleoside analogue that was approved by the FDA to target and inhibit the replication of novel and potentially re-emerging influenza viruses in 2014 in Japan [115]. Favipiravir targets the RdRp of RNA viruses, with specific activity against influenzas A, B and C, whilst also being able to inhibit rhinovirus replication in vitro [116]. Favipiravir binds to the RdRp’s active site, and it is mistaken as a purine nucleotide, which causes a confirmational change in its binding pocket, impacting its original function and, therefore, the chain is terminated [117]. It also induces C-to-U and G-to-A mutations in the RdRp active site, which are lethal and will disrupt viral replication in this way [117]. The structure of the antiviral molecule can be found in Figure 5.

Many clinical trials have tested the use of favipiravir in COVID-19 patients. A multicentre, open-labelled, randomised control study found that favipiravir may have significant clinical symptomatic improvements compared to the control group in mild to moderate COVID-19 patients over 5–14 days; however, it did not impact the viral load [120]. Those with a viral load at a lower baseline had greater viral reductions between 1 and 13 days during the favipiravir course [120]. However, other similar studies concluded that, even though early infection may increase the likelihood of ventilation-free survival in patients younger than 60 whilst being treated with favipiravir, there were ultimately no improvements in clinical outcomes [121]. Overall, this study found that there was not enough evidence to support the hypothesis of favipiravir being used for COVID-19 patients. An antiviral drug that can bind to and inhibit the replication of multiple CoVs is crucial to reduce the severity of inevitable future CoV emergences, as well as improving the likelihood of being used for other RNA viruses too.

Other small molecules can non-competitively bind to a viral protein involved in replication to interrupt its normal functions and indirectly prevent further replication [122]. They allosterically bind to a protein, away from the active site of an enzyme, to cause configurational changes within its 3D structure by forming or breaking hydrogen bonds/van der Waals interactions. This inhibits surrounding viral substrates from binding to the active site, and viral replication is, therefore, interrupted [122]. An example is nevirapine, which is a non-nucleoside reverse transcriptase inhibitor (NNRTI) and is commonly taken in combination with an NRTI such as ritonavir, a protease inhibitor that also inhibits the cytochrome P450 liver metabolism enzymes to prolong and increase the plasma concentration levels of the drugs [123,124]. The chemical structures of both inhibitors are presented in Figure 6.

Nevirapine is an HIV RT inhibitor, whilst ritonavir is an HIV protease inhibitor, with the capability of binding to the CYP liver enzymes for increased plasma concentrations. Studies have shown that patients who are HIV-positive and have a SARS-CoV-2 co-infection have reduced symptoms in both illnesses when taking these medications, therefore showing that the drugs can bind to both HIV and SARS-CoV-2 to prevent replication [126]. Both drugs have many interacting nitrogen electrophilic and oxygen nucleophilic sites, which can increase their binding reactivity [127]. Antivirals designed for SARS-CoV-2 will have specific protein targets, and the examples are discussed in Section 3.

## 3. Novel Viral Druggable Targets

### 3.1. Non-Structural Protein 3 (NSP3)

NSP3 is the largest protein encoded in all the CoV genomes, and it is a protease, which cleaves the polyprotein to release NSPs 1, 2 and 3. It is also an important component of the replication–transcription complex [41]. NSP3 is an interesting drug target as it is the next main protease after NSP5 and is the largest protein with many highly conserved amino acids across the different CoVs; however, only a small number of drug candidates have been designed, and none have reached clinical trials yet [128]. Peptidomimetic inhibitors called VIR250/VIR251 have shown evidence of good inhibition against NSP3 for both SARS-CoV and SARS-CoV-2 [129]. VIR250 interferes with the backbone–backbone H bonds with G271, and both VIR250 and VIR251 interfere with the H bonding in Y268, amino acids within the active site of NSP3 [129]. In cellular studies, VIR250/VIR251 was trialled and demonstrated to have high selectivity for both SARS-CoV and SARS-CoV-2 NSP3 and only slight inhibition in MERS-CoV. It builds a foundation for the development of a pan-CoV inhibitor; however, further research adaptations are needed [130]. Figure 7 presents the chemical structures of VIR250/VIR251.

There are currently no novel FDA-approved drugs that target the NSP3 of CoVs; however, there have been many studies finding that previously approved drugs, such as Ebselen, Ceftazidime and Thiomersal, have antiviral and inhibitory effects on SARS-CoV-2 NSP3, which could suggest coverage for other CoVs. However, due to other unknown effects the drugs could have for use outside of the original design, further studies are needed to determine this [131].

### 3.2. Non-Structural Protein 5 (NSP5)

All NSP5 proteases in CoVs identified are chymotrypsin cysteine proteases, and they are responsible for the cleavage of NSPs 4–16 (the majority of the polyprotein). This function is highly conserved across other CoVs, making it an attractive drug target [43]. NSP5 proteases generally have a sequence identity of >80% within the same genera, and other CoVs may have a similarity identity of ~50% across other genera; however, the greatest conservation point is the enzymatic active site [132]. Because of its similarity, drug targets against NSP5 may have an increased chance of retaining activity in other SARS-CoV-2 strains and possibly other HCoVs, improving the chances for its widespread use [133].

COVID-19 can now be treated with Paxlovid, which was approved by the Food and Drug Administration (FDA) for emergency use in December 2021 for the vulnerable population. It consists of nirmatrelvir, the novel antiviral that targets MPro (NSP5) of SARS-CoV-2, and ritonavir, an HIV-1/2 protease inhibitor that targets cytochrome P450 3A (CYP3A) enzymes to increase the concentrations of nirmatrelvir [134]. CYP enzymes are drug-metabolising enzymes, and, therefore, when they are inhibited, this prolongs a drug’s lifespan in the body by influencing drug–drug and drug–target interactions [135]. Nirmatrelvir is an analogue of GC373, a small molecule prodrug that was designed to target the MPro of feline and mink CoVs [136]. The adduct GC376, a bisulphate compound, is converted readily to the peptide aldehyde GC373 when it reaches its target [136]. The structure for nirmatrelvir is presented in Figure 8.

In a meta-analysis study identifying the efficiency of Paxlovid against COVID-19 in clinical trials, it was found that the drug has evidence of significantly decreasing the mortality rates of individuals infected with the virus compared to a healthy control group [138]. Paxlovid was also compared against molnupiravir and fluvoxamine, two other drugs that have been shown to reduce disease severity in COVID-19 patients, and Paxlovid was the best [138]. In a phase 2–3 double-blind, randomised control trial to identify the efficacy and safety of Paxlovid in unvaccinated, symptomatic adults with a high risk of developing severe COVID-19, the results showed that Paxlovid reduced the relative risk of disease severity by 89.1% in hospitalisation cases, after taking the medication every 12 h for 5 days within 3 days of symptom onset [133].

### 3.3. Non-Structural Protein 12 (NSP12)

Remdesivir is an intravenous drug, initially designed to target the NSP12 (RdRp) of Hepatitis C virus (HCV), and expanded to clinical trials for filoviruses, such as ebolavirus and Marburg virus, and, ultimately, SARS-CoV-2 [139]. Remdesivir is an adenosine NA and a phosphoramidate prodrug, which is metabolised to form remdesivir triphosphate, its active form [140]. Polymerases across different viruses share very similar functions and, therefore, have similar 3D structures and sequences, and because of this, pre-existing drugs targeting this enzyme are often repurposed to treat different viral infections [141].

Remdesivir generated results similar to molnupiravir in other studies, which is a prodrug of the β-D-N4-hydroxycytidine (NHC) nucleoside analogue that promotes C-U and G-A point mutations in the RdRp of SARS-CoV-2. It transcribes viral RNA using NHC triphosphate to cause these point mutations, resulting in non-functioning proteins, reducing viral activity [142,143,144]. It can be produced on a large scale and does not require specific temperature-regulated or hospital conditions for storage, and it was proven in clinical trials that there are no significant adverse effects of administering the drug to infected individuals [145]. Molnupiravir was originally designed to target the RdRp (NSP12) of influenza in 2019; however, NSP12 is a very versatile enzyme within RNA viruses, and its structural features that are core in its function are highly conserved across different viruses. Therefore, the drug could be repurposed for COVID-19 [145,146]. Figure 9 presents the chemical structures of both remdesivir and molnupiravir.

In a retrospective study, hospitalised patients with COVID-19 were treated with remdesivir (they were predominantly 40–60-year-old males who required oxygen therapy and were taking the treatment for 5 days) [149]. The infection improvement rate was reported in historical cohorts, as 84% and only 13% of the individuals in the study experienced adverse side effects [149]. In a randomised control trial, remdesivir and casirivimab/imdevimab were tested in low-risk symptomatic adult individuals to identify the viral clearance rate with the intention to treat the wider population with COVID-19 by investigating the viral load using PCR across 7 days [150]. The results showed that both therapeutics increased viral clearance in individuals with early COVID-19 infection because the viral burden is at its highest; however, there were uncertainties regarding the use of remdesivir and casirivimab/imdevimab (anti-spike mAb therapies) in hospitalised patients (once the infection is in the late stage, anti-inflammatory treatments are more effective) [150]. Effects were also dependent on the SARS-CoV-2 strain. Casirivimab was much less effective than imdevimab against the omicron variant that was circulating at the time, due to the N440K and G446S mutations in the Spike protein that reduce the activity of the drug in vitro [150]. Viral clearance rates increased by 42% during remdesivir treatment but only by 23% in the casirivimab/imdevimab group, indicating that remdesivir was more effective in reducing viral load in early COVID-19 infections [150]. A limitation of this approach is that intravenous monoclonal antibody and drug treatments are much more difficult to obtain, store and distribute across a large population, leading to increased costs and reductions in overall coverage. Therefore, it is likely more beneficial to opt for oral drug intake for non-hospitalised cases. Monoclonal antibody treatment such as casirivimab/imdevimab is also disadvantageous in targeting multiple different viral variants or genera and, therefore, is not ideal for being repurposed more widely for HCoVs [150].

An approach to decreasing efficacy could be to increase the dose and concentration of the mAb given, but this increases the risk of side effects and inflammation [151]. Developing an oral drug will improve the global coverage of treatment, accessibility and costs as well; therefore, mAbs may not be a feasible long-term treatment for CoVs [152,153]. NAs vastly improved the HIV pandemic by reducing viral loads to undetectable levels in numerous individuals and, therefore, have the potential to reduce SARS-CoV-2 transmission and infections with the existing HCoVs, as well as improving preparedness for the next inevitable HCoV outbreak [154].

There are numerous potential drug compounds that target NSPs 5 and 12 in available drug libraries. Other viral proteins may not be strong contenders because of their conservation levels across strains/genera. For example, the Spike protein has one of the highest mutation levels as it is constantly and spontaneously adapting to improve its binding to host cells, and we have seen mAb escape mutations. Others, such as NSP2, do not have properly understood functions in viral activity and replication, and, therefore, it becomes more difficult to identify core amino acids that can be targeted with drugs to reduce protein function [155].

## 4. Available and Alternative Treatments/Prophylaxis

Repurposing antivirals is an effective method to control outbreaks more rapidly, reducing costs in research and reducing the risk of failure or adverse effects [156]. There are many drugs that have been approved to be used for the treatment of COVID-19; however, most of them have been repurposed and were not designed for this use. The treatments that have been approved by the FDA include the following:Paxlovid (ritonavir and nirmatrelvir);Lagevrio (molnupiravir);Veklury (remdesivir)—this is only approved for use in individuals who are at high risk of developing severe COVID-19 infection, including the vulnerable/immunocompromised population.

More details on each of these drugs can be found in Section 3.

There are various monoclonal antibodies (mAbs) that block viral entry into human cells and neutralise the virus before being able to infect. These include the following:Sotrovimab;Bebtelovimab;Casirivimab/imdevimab.

Although mAbs are an effective treatment, as they target the Spike protein to prevent viral entry into human cells, there is a significant risk of escape mutants developing as the Spike gene has a very high mutation rate. Therefore, mAbs can become ineffective against new strains and variants [157,158]. One approach to avoid this problem is to use mAb cocktails to reduce the risk of escape mutants. Mutations such as N439K and Y453F within the S protein are proven to reduce the efficiency of mAb treatment [151]. These mutations have been found in SARS-CoV-2 Spike sequences in >30 countries since January 2021, and they provide improved affinity between RBD and the ACE2 receptor, leading to increased viral loads compared to the ancestral Wuhan CoV [159].

There are many antiviral compounds being tested in clinical trials for COVID-19, such as Paxlovid, one of the first drugs designed specifically to target SARS-CoV-2, and each will bind to a particular protein that has crucial functions in viral replication. Examples are detailed below.

Anti-malarials such as chloroquines were tested during the COVID-19 pandemic because of their interference with ACE2 glycosylation [160]. The drug is cheap and widely available, yet chloroquine resistance is already a significant public health problem [161]. Studies have shown that chloroquines may be able to inhibit SARS-CoV-2 replication by increasing the endosomal pH and denaturing viral enzymes, which require lower pH levels; however, this study has methodological limitations, including the lack of a control group and unknown interventions, and more clinical data were necessary [162]. A clinical study showed that the odds ratio of mortality rates in SARS-CoV-2-infected individuals taking hydroxychloroquine and chloroquine were 1.11 and 1.77, respectively, indicating an association of increased mortality rates in SARS-CoV-2 cases with taking chloroquines, indicating that there were no molecular benefits [163]. Many drugs were trialled against SARS-CoV-2 during the pandemic due to the emergency nature; however, the challenge remains to design antivirals that do not interfere with host cell functions and target viruses specifically. This is of the utmost importance and an important research area to target for future epidemics, pandemics and novel infectious outbreaks [164].

## 5. The Design of a Novel Antiviral

It is crucial to have a small molecule drug readily available in emergency situations that has the capability to target multiple viruses, especially for future inevitable emerging HCoVs. The repurposing of drugs is also important to save on cost and time and to improve our preparedness for the next outbreak to reduce mortality and hospitalisation rates.

During the COVID-19 pandemic, studies showed that between January 2021 and August 2022, the monthly rates of SARS-CoV-2-related hospitalisations were 5.2× greater in unvaccinated individuals aged > 18 compared to those vaccinated, with general cases at the lowest rate within the vaccinated group who also received a booster [165,166]. However, as the strains mutate the Spike sequences and generate dominant variants, the vaccines become less effective because of decreased specificity and selection for viral escape. Therefore, designing an antiviral drug within a more conserved region of the CoVs will improve the chances of binding to multiple current and future CoVs, further improving our preparedness and control efforts. Sufficient treatments are needed due to the emergence of three severe CoVs over the last 20 years and no prior general antivirals being available. Targeting multiple HCoVs, including future emerging HCoVs, will, thus, be important in limiting infections, hospitalisations, deaths and disease severity. Reduced transmission rates will lead to fewer mutations and virus evolution, resulting in better control [167].

An approach for the design and synthesis of novel small molecules with the capability of targeting multiple HCoVs includes the identification of proteins with functions critical for viral replication, as they will often have a lower mutation rate, since mutations in active or enzymatic sites would likely negatively hinder that specific function [168]. These regions will, therefore, be more conserved across different HCoVs, making these an attractive drug target [169,170]. The challenge is finding the optimal target and designing a small molecule inhibitor that has broad CoV activity. Researching all seven HCoVs and identifying the roles of each protein in replication are first required. Subsequently, a multiple sequence alignment (MSA) will help identify the most conserved proteomes and their active sites compared to those identified using the available 3D PDB structures. Examples of conserved NSPs include NSP3 and NSP5, two different proteases, and NSP12, the RdRp [128]. A lead compound must be identified from PDB, and in in silico studies, using a ligand-bound structure will help to locate the active site of the protein, and the design of the small molecules, which will bind to these active sites, can take place in vitro. Cellular and molecular assays using different CoVs (CoV-NL63, CoV-229E, CoV-OC43, CoV-HKU1, SARS-CoV, MERS-CoV and SARS-CoV-2) will determine whether these novel molecules will prevent viral infection and the cytopathic effect (CPE) in cell lines, along with other confirmatory assays to measure cytotoxicity and specific interruptions to viral replication [171,172]. If successful, these newly designed drugs will have the capability of broadly targeting more than one HCoV, including those yet to emerge, which will be hugely impactful for future CoV outbreaks. The ultimate aim is to help prevent the detrimental impacts to public health.

## 6. Conclusions

Future coronavirus outbreaks are inevitable, as zoonotic diseases cannot simply be eradicated; therefore, the preparation, prevention and prophylaxis are key for their control [173]. CoVs have existed for at least ~70 years, most likely much longer, yet the first successful antiviral, Paxlovid, was approved by the FDA for emergency use in 2021 [174]. There have been many small molecule compounds in clinical trials to reduce the mortality and hospitalisation rates of COVID-19 patients, such as various mAb treatment, vaccines and other small molecule compounds, all of which seem to focus solely on SARS-CoV-2. Targets, such as NSP3, NSP5 and NSP12, have highly conserved binding sites, which make them attractive drug targets, and there has been minimal research into how trialled drugs will affect both mild and severe CoVs, but COVID-19 always remained the prime focus [168]. Vaccines and mAbs typically target the Spike protein, an area with a very high mutation rate and significant differences between the existing CoVs, and, therefore, they would only be most successful in the particular target strain and not for mutant or future CoVs [151,175]. The novel approach proposed here uses a bioinformatic approach to identify conserved viral proteins and their functions, then using the PDB to screen for potential lead compounds. Analogues are designed based on the identified lead compound and synthesised in vitro to be tested in cellular and molecular assays in vitro to first evaluate antiviral effects. Other confirmatory assays will generate solid evidence of a novel compound that can interrupt the viral replication of mild CoVs in vitro. Ultimately, the data collected should be extrapolated for use in current (SARS, MERS, SARS-CoV-2) and future severe CoVs such as SARS-X.

## Figures and Tables

**Figure 1 viruses-16-00156-f001:**
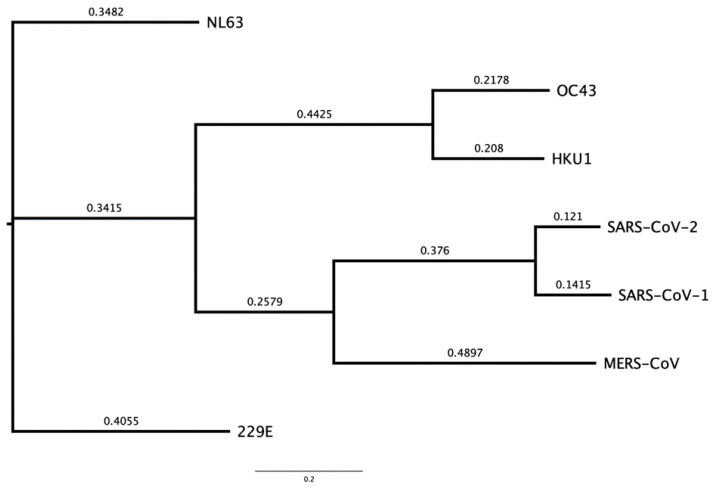
A phylogenetic tree of all 7 HCoVs including a calculated distance between branches and nodes in nucleotides substitution per site [28,30]. The genetic sequences and reference genetic codes were obtained from the National Centre for Biotechnology Information (NCBI) [31]. The nodes represent the most recent common ancestor of the lineages, and the values are the measure of support for each node, the values are between 0–1 and the higher the value the stronger the evidence that the sequences cluster together this way [32]. Accession codes: CoV-NL63–NC_005831, CoV-229E–NC_002645, CoV-HKU1–NC_006577, CoV-OC43–NC_006213, SARS-CoV–NC004718, MERS-CoV–NC_019843 and SARS-CoV-2–MT786327.

**Figure 2 viruses-16-00156-f002:**
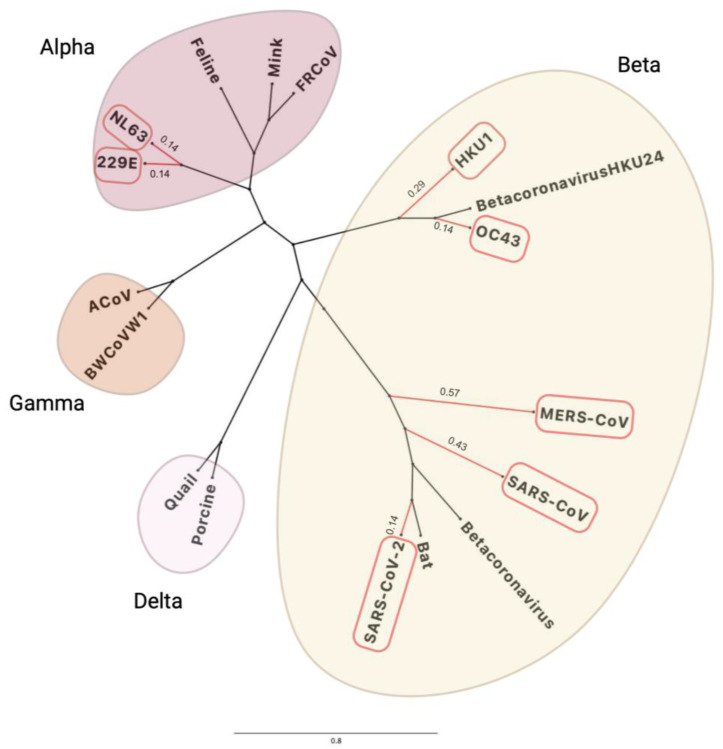
The taxonomic groups of human and zoonotic CoVs which include alpha, beta, gamma and delta groups created using FigTree and BioRender, also including a calculated distance between branches and nodes in nucleotides substitution per site [29,35]. Those with branches highlighted in red are the HCoVs also presented in Figure 1. The accession codes for the CoV genomes were found using NCBI (MN996532.2 (Bat CoV RaTG13), MZ081381.1 (RpYN06 betacoronavirus), MZ802777.1 (porcine deltacoronavirus), MH532440.1 (quail deltacoronavirus), MK841495.1 (porcine endemic diarrhea virus), MN535737.1 (mink coronavirus 1), KM347965.1 (FRCoV-NL-2010), NC_002306.3 (feline infectious peritonitis virus), NC_026011.1 (HKU24 betacoronavirus), MZ368698.1 (avian coronavirus), NC_010646.1 (beluga whale coronavirus SW1), and the HCoVs accession codes are in Figure 1.

**Figure 3 viruses-16-00156-f003:**
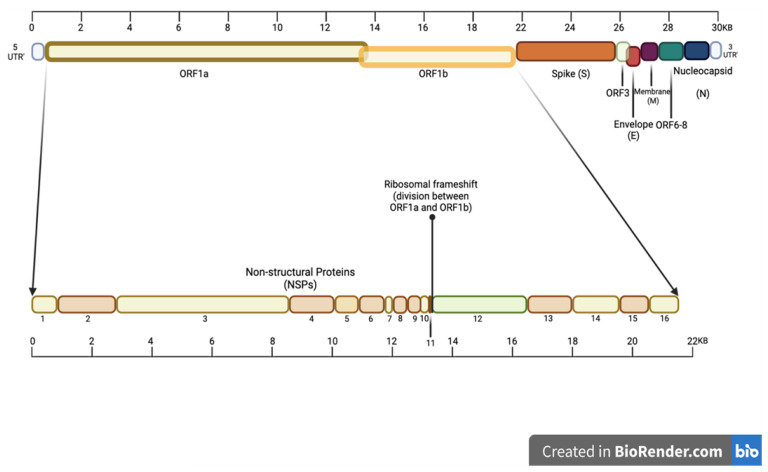
The SARS-CoV-2 genome divided into its NSPs, ORFs, and notable structural proteins such as the spike (S), envelope (E), membrane (M) and nucleocapsid (N) [36]. Other important genes include ORFs 3 and 6–8 at the 3′ UTR. Diagram inspired by Ellis et al. 2021 and created using BioRender [35,36].

**Figure 4 viruses-16-00156-f004:**
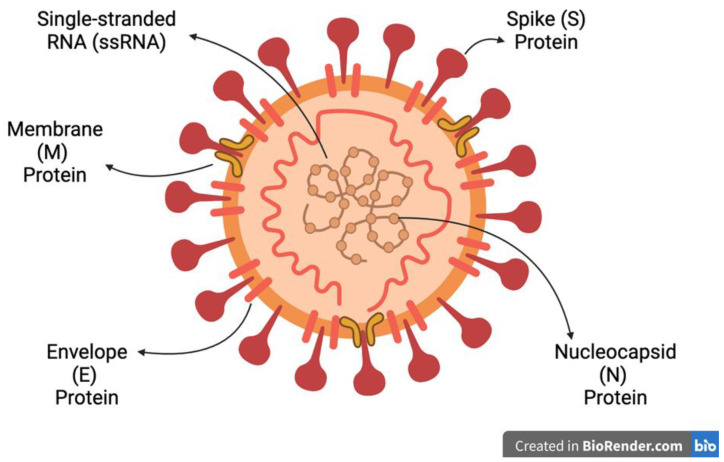
A visual presentation of a generic human CoV virion, with image inspiration from Boopathi, S and Afzal, A and created in BioRender [35,71,72].

**Figure 5 viruses-16-00156-f005:**
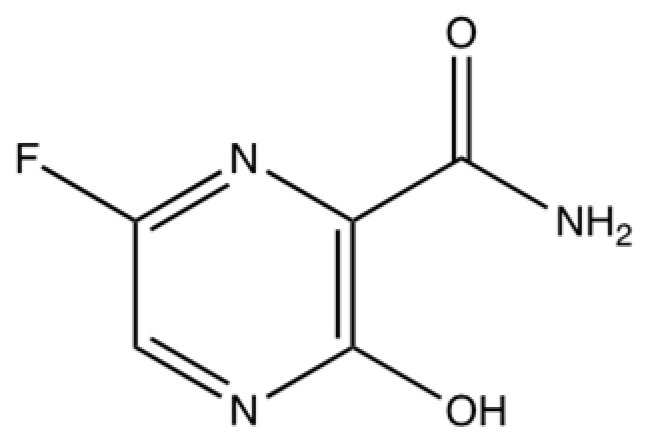
The chemical structure of favipiravir [118,119].

**Figure 6 viruses-16-00156-f006:**
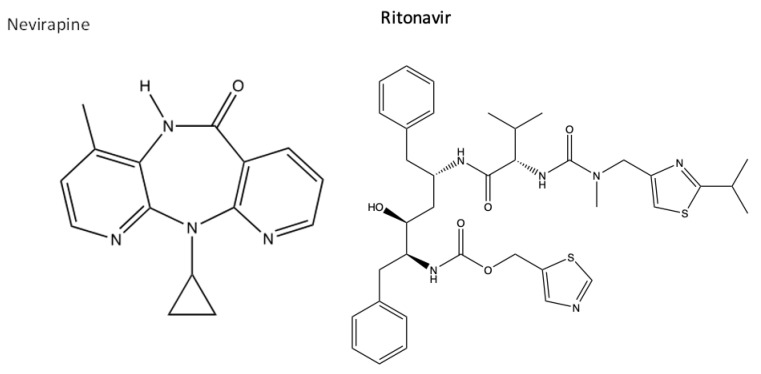
The chemical structures of nevirapine and ritonavir [119,125].

**Figure 7 viruses-16-00156-f007:**
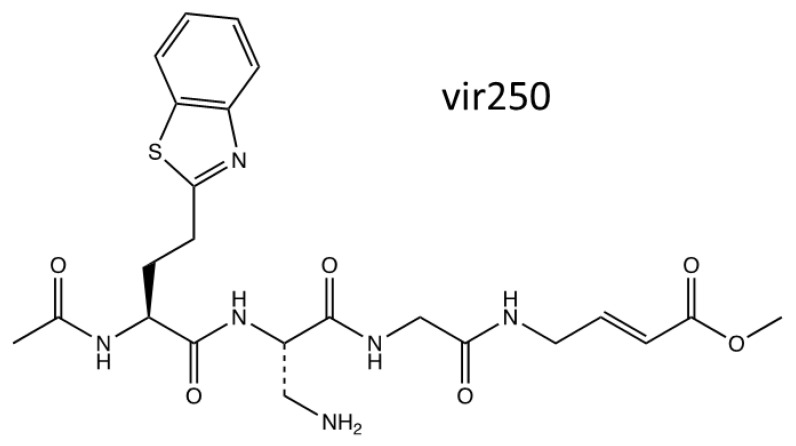
The chemical structures for VIR250/VIR251 [119,130].

**Figure 8 viruses-16-00156-f008:**
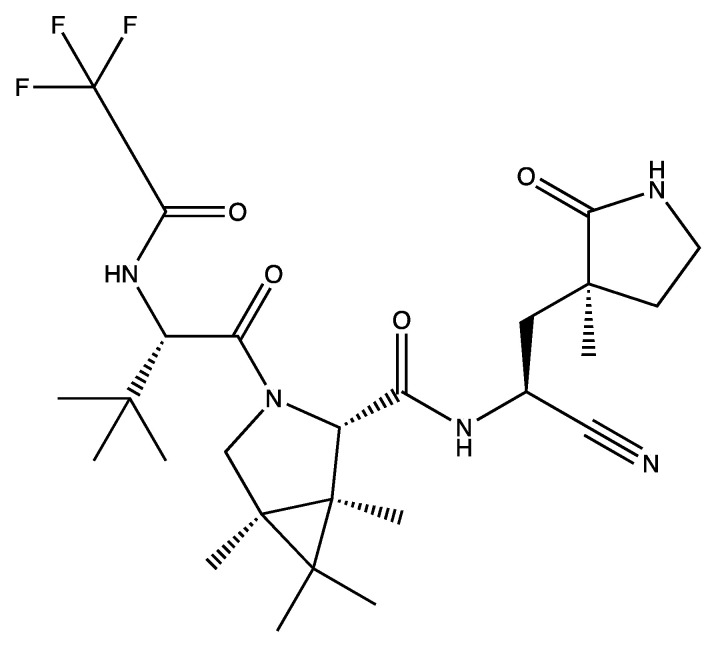
The chemical structure of nirmatrelvir [119,137].

**Figure 9 viruses-16-00156-f009:**
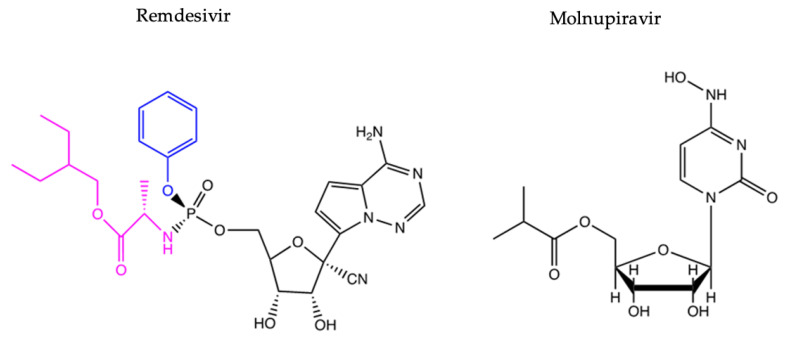
The chemical structures for remdesivir and molnupiravir [119,147,148].

## Data Availability

Data contained within are from source reference material, and the authors can provide additional data following reasonable written request.

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
