# Peer review of "Antivirals for Broader Coverage against Human Coronaviruses"

_viruses, 2024, doi:10.3390/v16010156_

Round 1

Reviewer 1 Report

Comments and Suggestions for Authors

The manuscript titeled “Antivirals and Biologics for Broader Coverage Against Human Coronaviruses” by Outteridge and co-workers is a well written, timely and comprehensive review about the current state of antiviral research and drug development to combat replication and transmission of human pathogenic coronaviruses.

The authors started with an insightful introduction, providing a historical perspective on human coronaviruses and elucidating their genetic diversity. The subsequent section of the review navigates through the treatment options for HCoV infections, offering a summary of currently FDA-approved drugs. The final segment concentrates on innovative viral drug targets, alternative treatment options, and vaccine strategies. Finally, the authors give a short outlook about the design of novel antivirals.

Following the recommended modifications, the review is deemed suitable for publication in "Viruses." However, a few enhancements are suggested:

  1. Figure 2, page 4:
    • In the legend, elucidate the significance of numbers (e.g., 0.14 relative distance to…).
    • Address issues where lines and numbers (e.g., 0.14) overlap.
    • Consider incorporating visual enhancements to improve the clarity of the figure.
  2. Table 1:
    • Condense the table, spanning from page 5 to page 10, to a more compact version for improved readability.
  3. Include a Short Chapter:
    • Introduce a brief chapter summarizing the current status of host-targeting antivirals.
  4. Chapters 6.0 and 7.0:
    • Augment the number of references in these chapters to bolster the scholarly foundation of the content.
Comments on the Quality of English Language

No comments

Reviewer 2 Report

Comments and Suggestions for Authors

see attach

Round 2

Reviewer 2 Report

Comments and Suggestions for Authors

the authors addressed the various comments ; manuscript can be accepted 

line 539 : the word 'however' is  mentioned twice